# Sarcoma Predisposition in Dogs with a Comparative View to Human Orthologous Disease

**DOI:** 10.3390/vetsci10070476

**Published:** 2023-07-21

**Authors:** Maja L. Arendt, Jane M. Dobson

**Affiliations:** 1Department of Veterinary Clinical Sciences, University of Copenhagen, 1870 Frederiksberg C, Denmark; 2Queens Veterinary School Hospital, University of Cambridge, Cambridge CB3 0ES, UK

**Keywords:** genetic risk factors, sarcoma, canine

## Abstract

**Simple Summary:**

Sarcomas are tumors arising from soft tissue or bone that occur in both humans and dogs with varying frequency. Recently hereditary risk factors related to sarcoma development have been investigated in both species. This review highlights some of the recent findings relating to hereditary risk factors for sarcoma development in humans, and summarizes recent studies investigating hereditary risk factors for sarcoma development in dogs.

**Abstract:**

Sarcomas are malignant tumors arising from the embryonic mesodermal cell lineage. This group of cancers covers a heterogenous set of solid tumors arising from soft tissues or bone. Many features such as histology, biological behavior and molecular characteristics are shared between sarcomas in humans and dogs, suggesting that human sarcoma research can be informative for canine disease, and that dogs with sarcomas can serve as relevant translational cancer models, to aid in the understanding of human disease and cancer biology. In the present paper, risk factors for the development of sarcoma in dogs are reviewed, with a particular focus on recent advances in clinical genetics, and on the identification of simple and complex genetic risk factors with a comparison with what has been found in human orthologous disease.

## 1. Introduction

More than 70 sarcoma types, arising from soft tissues or bone, have been described in humans. The World Health Organization recently published the 5th edition of their classification system for human sarcomas, which, in addition to histopathological and immunohistochemical criteria, includes molecular and genetic criteria for more precise classification and treatment guidance [1]. Many histological, biological and genetic features are shared between sarcomas in humans and dogs. However, molecular classification schemes have not yet been adapted to dogs, although there is evidence to suggest that many of the same genes are mutated or differentially expressed in orthologous sarcomas between humans and canines [2,3]. In addition, many of the same sarcoma types exist in both species, though with varying frequency [4]. 

Sarcomas in humans are generally considered rare, accounting for only 1% of cancers in adults. In children, however, sarcomas account for a larger fraction of cancers, with around 20% of pediatric cancers being sarcomas [5,6]. An epidemiological study investigating the incidence of sarcomas in the US between 2002 and 2014 showed that the overall incidence in the period was 7.1/100,000 people. However, when considering individual years, there was a trend showing an increase in incidence during the study period, with the incidence rising from 6.8/100,000 people in 2002 to 7.8/100,000 people in 2014 [7]. Due to the rarity of the disease, and the many specific sarcoma types, human studies commonly group sarcomas together to increase the study’s size. In contrast, canine studies tend to divide sarcomas into soft tissue sarcomas, commonly including non-oral fibrosarcoma, peripheral nerve sheath tumors, and perivascular wall tumors, and sarcomas classified according to their cell of origin and tissue location. This division reflects the biological behavior of the tumor type. Sarcomas are more common in dogs than in humans. A recent German study evaluating the histopathological diagnosis of 70,966 benign and malignant canine tumors showed that fibrosarcomas alone accounted for 5.8% of the neoplastic lesions; however, when including other sarcoma types, the overall percentage of sarcomas was closer to 13%, though the lack of histological details in the study does not allow for a precise estimation. If only comparing these against malignant tumors in this database, the fraction would likely be substantially larger [8]. A review summarizing data on sarcoma occurrence in the United States showed that in the four studies reviewed, sarcomas accounted for 11.7–16.4% of canine cancers [4]. A study from the United Kingdom (UK) calculated the yearly incidence of different tumors based on insurance data from dogs. The yearly incidence of soft tissue sarcomas in this study was 122/100,000 dogs, whilst the incidence for osteosarcoma was 57/100,000 dogs. The incidence for hemangiosarcoma in this study was 24/100,000 dogs, whilst the incidence for cardiac hemangiosarcoma was 1/100,000 dogs [9]. It is difficult to directly compare human and canine epidemiological data due to differences in disease surveillance and the accelerated life span of dogs; however, overall, sarcomas account for a larger fraction of malignant tumors and occur with a substantially higher yearly incidence in dogs compared to humans. 

## 2. Environmental Risk Factors for Sarcoma Development in Humans and Dogs

Several studies have investigated environmental risk factors associated with sarcomas in humans and dogs, with variable evidence supporting associations [5,10,11]. Ionizing radiation has consistently been observed as a common risk factor particularly for osteosarcoma development in humans and dogs. This supports the notion that the genetic damage induced to normal tissues during radiation therapy predisposes patients to secondary sarcoma development [12,13,14]. Another risk factor, which has been observed in humans, is infection with human herpes virus type 8 (HHV8) [5,10,15]. In dogs, unlike cats and humans, there is no evidence to suggest that viruses play a role in sarcoma development. Other risk factors proposed in humans with lesser evidence are occupational exposure and exposure to different chemicals (such as phenoxyherbicides or chlorophenols), trauma, immunosuppression, increased bodyweight and lymphedema [10,16,17,18,19]. An interesting study looking at cancer risk in Finnish athletes showed that hurdlers had a statistically significantly higher risk of soft tissue and bone sarcomas [20]. Studies in dogs have shown that neutering at a young age increases the risk of developing osteosarcoma in both males and females, indicating a putative role for hormone regulation during adolescence [11,21]. Other risk factors in dogs include size and weight [22,23]. However, due to the inheritance of these traits, it can be difficult to separate these from genetic risk factors. Interestingly, in a study looking at risk factors for osteosarcoma in dogs, all greyhounds affected by osteosarcoma were racing greyhounds [14]. The observation of sarcomas in both racing greyhounds and human hurdlers suggests that certain activities could be linked to sarcoma development, though the biology behind this is not completely understood [14,20]. Whilst environmental risk factors associated with sarcoma development have been described in humans and dogs, the observation of families or dog breeds with an increased frequency of sarcoma development has led to increasing attention being devoted to predisposing genetic risk factors.

## 3. Genetic Risk Factors for Sarcoma Development in Humans and Dogs

Genetic risk factors have received a lot of focus in human sarcoma research. Differences in incidence between ethnic groups and observations that sarcomas are more frequent within certain families indicate an underlying genetic cause for a subset of sarcomas [24,25]. Furthermore, sarcomas account for a high proportion of cancers in young children, which have had limited exposure time to environmental risk factors, which again suggests that germline genetic variation plays a role in the development of this cancer type [10]. Similarly, it has been shown that there is a marked difference in the risk of developing sarcomas between different dog breeds [26,27]. Studies have also investigated pedigrees showing a high proportion of dogs developing specific sarcoma types, or multiple different cancer types including sarcomas within a family tree, which, in a human setting, would have been in alignment with the Chompret criteria, used for identifying individuals to be screened for Li Fraumeni syndrome [28,29,30]. With the recent advances within genetic research facilitated by next-generation sequencing, it has been possible to investigate and identify genetic risk factors for diseases with simple monogenic or complex poly- genic inheritance. 

## 4. Genetic Predisposition to Sarcomas in Humans

Genome-wide association studies (GWAS) have been performed in humans, identifying susceptibility loci associated with the development of osteosarcoma and Ewings sarcoma [31,32]. However, recent research has focused on identifying rare genetic variants affecting disease development. Hereditary cancer syndromes are commonly described as autosomal dominant Mendelian inherited syndromes, caused by genetic variants with variable degrees of penetrance, though other types of inheritance patterns exist. This means that for most syndromes, not all carriers will be affected by cancer despite carrying genetic variants which have debilitating effects on the function of a particular gene. At least 22 familial cancer syndromes leading to predisposition to sarcomas in humans have been described [25,33]. These syndromes are most commonly caused by having an inactivating mutation in genes with tumor-suppressing functions, such as *TP53*, which predisposes patients to multiple sarcoma types, *NF1,* which predisposes patients to neurofibrosarcoma, and *RB1,* which predisposes patients to osteosarcoma as well as soft tissue sarcomas [25]. In humans, there are guidelines for when to suspect that an individual could be affected by a hereditary cancer syndrome, such as having several relatives developing cancer, developing cancer at a particular young age or developing multiple cancer types simultaneously or sequentially [28,29]. It is evident that most cancer syndromes can manifest as a plethora of different cancer types. With the addition of the incomplete penetrance of the phenotype, it can sometimes become difficult to recognize that several different cancers within a family share the same predisposing genetic risk factors [25]. Due to the increased frequency of sarcomas in hereditary cancer syndromes in humans and increased incidence within certain populations, several recent research studies have investigated the presence of germline genetic risk variants in humans affected by sarcoma without selecting for individuals with a family history of cancer. Ballinger et al. investigated germline genetic risk factors in 1162 sarcoma patients by performing exome sequencing of 72 genes known to have a role in cancer predisposition [34]. In the study, they found that 55% of sarcoma patients carried an excess of pathogenic germline variants compared to a normal control population, and that these pathogenic variants were associated with an earlier age of onset of the disease. The study indicates that more than half of sarcoma patients carry putative monogenic or polygenic risk factors in known cancer genes [25,34]. In a more recent study by the same author, whole genome sequencing was used to identify heritable predisposing genetic variants in 1644 sporadic sarcoma patients and a comparison group of 3205 healthy geriatric controls, allowing for a more detailed analysis of genetic variants across the whole genome. The study showed that 6.6% of the sarcoma patients carried either known pathogenic, likely pathogenic or novel loss of function variants in genes known to be relevant to cancer predisposition, such as *TP53*, *ATM*, *BRCA2*, *BRCA1* and *CDKN2B* [35]. Again, carriers of these variants were more likely to develop sarcomas at an earlier age and to develop multiple primary cancers. An extended analysis was performed including genetic variants with unknown significance, classified as possibly pathogenic according to the American College of Medical Genetics guidelines [36]. This analysis, with stringent comparison to control groups, identified 968 genes with increased burden of possibly pathogenic variants in the sarcoma group [35]. Strikingly, the data showed that 3.2% of the sarcoma patients in the study carried variants related to the shelterin complex and telomere genes, with only 0.2% of the reference control population carrying variants in these genes. This highlights altered telomeric maintenance as an important pathway predisposing patients to sarcomas in humans [35]. Other studies have focused on specific sarcoma types, such as the study by Mirabello et al., which showed that 28% of osteosarcoma patients carried pathogenic or likely pathogenic genetic variants affecting genes known to be associated with cancer susceptibility. They found that the variants affecting *TP53*, *CDKN2A* and *BRCA2* were the most common. Interestingly, they also showed that telomer- and shelterin-associated genes *POT1*, *TERT* and *TINF2* were among the genes affected by pathogenic or likely pathogenic variants [37]. Additional studies investigating genetic variants in both adult and pediatric sarcoma cohorts as well as specific sarcoma subtypes all confirm that germline genetic predisposition plays an important role in the development of sarcomas, with between 6.6 and 28% carrying pathogenic or likely pathogenic variants in cancer-susceptibility genes [37,38,39,40]. 

## 5. Identification of Predisposing Risk Factors for Sarcoma Development in Dogs

Due to a high level of disease segregation for specific sarcoma types within particular dog breeds, some sarcoma types are considered to be heritable, though the exact mode of inheritance has not been proven. However, both simple Mendelian and complex inheritance patterns have been proposed [30,41,42]. Only a single familial cancer syndrome, caused by a genetic alteration in the FLNC gene, which is not linked to sarcoma development, has been described in dogs [43]. However, clear differences in breed predisposition suggest that a substantial fraction of disease risk for particular sarcoma types can be explained by genetics [30,41,43]. Research studies over the past 10–15 years have sought to identify hereditary patterns of disease, and identify mono- or polygenic risk factors for specific sarcoma types. Multiple GWASs have been performed to identify genetic loci associated with the development of specific sarcoma types in dogs [44,45,46,47,48,49]. Major findings from these studies are summarized in Table 1. A couple of studies have also used whole exome or whole genome sequencing to identify germline genetic variants in genes which could be related to an increased risk of developing cancer [2,50]. 

Osteosarcoma: Two GWASs have investigated the genetic risk factors associated with osteosarcoma in four different dog breeds with a high risk for developing osteosarcoma [47,48]. Karlsson et al. investigated the genetic risk factors in Rottweilers, Greyhounds and Irish wolfhounds by performing a GWAS comparing cases affected by osteosarcoma with healthy controls [47]. The study identified 33 nominally significant loci, of which a subset of 9 loci were defined as being genome wide significant by permutation testing (1 locus in greyhounds, 6 loci in Rottweilers and 2 loci in Irish wolfhounds). Interestingly, a very high proportion of Rottweilers and Irish wolfhounds carried the risk allele marking the associated greyhound locus, with allele frequencies in both cases and controls between 0.92 and 0.97 meaning that this variant is nearly fixed in the tested population. The locus defining SNP was further evaluated in cases and controls from other dog breeds, namely Leonberger and great Pyrenees dogs, with an increased risk allele frequency identified in these cases compared to that of the controls. The SNP marks a locus upstream of the *CDKN2A* and *CDKN2B* tumor suppressor genes, suggesting that an altered regulation of the expression of these genes could be predisposing dogs to the disease. An independent GWAS, published by Letko et al., investigating genetic risk factors associated with osteosarcoma in Leonberger dogs identified one significantly associated locus overlapping the previously identified risk locus in greyhounds [48]. This second GWAS emphasizes the association of the *CDKN2A*/*CDKN2B* locus with osteosarcoma development in large and giant dog breeds. 

Two studies investigated the mutational landscape of osteosarcoma in dogs based on whole exome and whole genome sequencing of tumor and normal DNA. Due to known hereditary cancer syndromes associated with osteosarcoma in humans, and earlier GWASs in dogs identifying genetic loci associated with osteosarcoma development, both studies also reported the presence of germline genetic variants for a selected set of genes in addition to the somatic tumor mutations [2,50]. Sakthikumar et al. performed a study looking at somatic mutations using whole exome sequencing of osteosarcoma DNA and matched normal control DNA in three high risk breeds, namely golden retriever, Rottweiler and greyhound. The study reported the presence of germline genetic variants affecting a select set of genes. These included genes previously associated with osteosarcoma in canine GWASs (*CDKN2A*, *CDKN2B*), genes associated with osteosarcoma based on a human GWAS (*GRM4*, *NFIB*) and genes associated with osteosarcoma in known human hereditary cancer syndromes (*BLM, RB1, TP53* and *WRN*) [32,33,47]. The study showed that 59/66 dogs carried genetic variants with a predicted moderate or high impact on gene function based on annotations carried out via variant effect prediction (VEP) [2,51]. Gardner et al. investigated somatic mutations in osteosarcoma by performing whole genome sequencing (*n* = 24) and whole exome sequencing (*n* = 13) of DNA from tumor tissue and matched normal DNA from the same dogs. In addition to somatic tumor mutations, the study reported the presence of germline genetic variants affecting a total of 28 genes, with previous implications in either canine or human osteosarcoma [2,32,47,52]. The data showed that all sequenced individuals carried at least one germline variant affecting the genes in the selected gene list, with a predicted moderate-to-high functional impact [50]. Interestingly, both studies reported a high frequency (30–41%) of individuals carrying mutations affecting the *CDKN2A* and *CDKN2B* genes previously associated with osteosarcoma risk. Both studies report that hereditary consequential variants are found in dogs with osteosarcoma, which suggests that osteosarcoma predisposition could be caused by simple hereditary risk factors similar to some of the recognized human sarcoma syndromes. We do, however, lack a comparison to a large cohort of healthy individuals in order to determine whether the identified variants are unique or occur at a higher frequency in dogs with a high risk of developing sarcomas. Further, we lack classification of coding variants in genes, similar to what is used in human genetics, to be able to interpret their pathogenic potential [35,53]. These variants could be transposed to the human genome to evaluate whether they overlap with confirmed pathogenic human variants in respective genes as well as to aid in the interpretation of their effect. A recent study investigating *BRCA1/2* variants in dogs with mammary tumors showed that none of the identified *BRAC1/2* variants overlapped with known pathogenic predisposing human variants [54]. Further prospective studies investigating germline mutations in breeds with a high risk of developing sarcomas are necessary to understand their role. Should a subset of these variants be shown to be dominantly inherited factors with incomplete penetrance, leading to an increased cancer risk, these variants could potentially be useful markers for selective breeding and screening programs, in addition to emphasizing dogs as comparative models for clinical studies in hereditary cancer syndromes. 

Hemangiosarcoma: Hemangiosarcoma has been reported at a high frequency in American golden retrievers, with a survey reporting a lifetime risk of 18.7%, based on a population of 427 deceased golden retrievers [55]. The same survey also reported that more than 50% of golden retrievers died due to neoplasia. The high frequency of neoplasia, and in particular hemangiosarcoma, suggest that underlying genetic risk factors could play a role in disease risk. A GWAS published by Tonomura et al. investigated the genetic risk factors associated with hemangiosarcoma and diffuse large b-cell lymphoma in American golden retrievers by comparing the genetic markers across the genome between cases and controls, and accounting for complex relatedness between individuals. The study found two independent genome-wide significantly associated loci on chromosome 5, as well as nominally significant loci on other chromosomes [46]. Though the study identified risk haplotypes associated with the development of hemangiosarcoma and b-cell lymphoma, no causal variants were pointed out as driving the predisposition. However, an expression study performed on tumor tissues from individuals carrying different haplotypes identified differences in the expression of nearby genes. Ideally, to interpret the effect of germline risk factors on differences in gene expression, experiments should include normal tissue expression as the tumor tissues carry additional somatic mutations, which affect gene expression. No subsequent studies have prospectively explored the association between the identified risk haplotypes and the development of hemangiosarcoma and b-cell lymphoma in golden retrievers. However, subsequent publications related to other dog breeds have identified an overlapping locus associated with histiocytic sarcoma [38,49]. Similar to osteosarcoma, multiple studies have evaluated somatic tumor mutations in hemangiosarcoma tissues using next-generation sequencing comparing data from matched tumor and normal tissue from the same individuals. Though data accounting for germline variation are available, putative germline variants leading to a potential risk of developing hemangiosarcoma have not been investigated in these studies [56,57,58].

Histiocytic sarcoma: Histiocytic sarcoma is a rare cancer type. However, in certain dog breeds, this cancer occurs at a high frequency, with studies reporting 20–25% of Bernese mountain dogs and flat-coated retrievers being affected by the disease [41,59,60]. Three GWASs have investigated genetic associations with histiocytic sarcoma development, either alone or in combination with other cancer types, in selected dog breeds. Shearin et al. performed a GWAS in Bernese mountain dogs originating from Europe and the US. The study identified two significantly associated loci on chromosomes 11 and 14, with the European population showing the strongest association with the chromosome 14 locus, whilst the American population and the combination of the American and European data showed the strongest association with the chromosome 11 locus [44]. Interestingly, the chromosome 11 locus identified is located around the *MTAP* and *CDKN2A/CDKN2B* genes and overlaps with the *CDKN2A/CDKN2B* locus, which is associated with osteosarcoma in other GWASs [44,47,48]. Hedan et al. published a GWAS study investigating genetic risk factors for histiocytic sarcoma in Bernese mountain dogs in combination with other predisposed breeds, namely flat-coated retrievers, Rottweilers and golden retrievers. The study also included dogs affected by lymphoma or mast cell tumors, as all three cancer types are common in Bernese mountain dogs [41,45]. The study identified a locus on chromosome 11 as being the most strongly associated, with additional loci identified on chromosomes 2, 5, 14 and chromosome 20 [45]. Again, the chromosome 11 locus identified overlaps with the previously identified locus on chromosome 11, which is associated with histiocytic sarcoma in Bernese mountain dogs, and osteosarcoma in other breeds. This confirms previous findings, and emphasizes the role of the locus in cancer predisposition. Interestingly, the locus identified on chromosome 14 overlaps with the POT1 and POT1-AS1 genes, suggesting that genetic variants affecting genes related to telomeric maintenance play a role in sarcoma predisposition in dogs as well as in humans. Evans et al. published a GWAS investigating the genetic predisposition to histiocytic sarcoma in flat-coated retrievers [49]. The study identified a locus on chromosome 5 as being significantly associated with development of disease, with a second locus being identified on chromosome 19. The gene *PIK3R6*, located in the chromosome 5 locus, showed allele-specific differential expression. However, the exact causative variant related to the expression was not identified. This locus overlaps with the locus previously described as associated with hemangiosarcoma in golden retrievers, as well as the chromosome 5 locus in the multibreed GWAS for histiocytic sarcoma, which also included flat-coated retrievers, as mentioned above [45,46]. Though all three of the abovementioned GWASs included sequencing of the risk loci for a subset of individuals, none of the studies identified clear causal functional variants, though many candidate variants and associated haplotypes were presented. 

## 6. Conclusions

Current research has identified genetic risk loci associated with sarcoma development in dogs, with a locus on chromosome 11 identified to be associated with the risk of osteosarcoma and histiocytic sarcoma, and loci on chromosome 5 associated with the risk of developing hemangiosaroma and histiocytic sarcoma in retrievers, as well as additional risk loci being reported. The overlap of associated loci between different studies is striking and emphasizes the importance of these loci and their role in cancer predisposition in dogs. Furthermore, these findings show that risk loci can be shared across breeds and cancer types. Additional research is needed to identify casual variants in these loci and understand their functional impact on cancer development. In humans, the identification of rare genetic variants accounting for the risk of sarcoma development have been made possible by the next-generation sequencing of large cohorts. This approach has yet to be undertaken in the canine population, though studies in osteosarcoma have investigated variants affecting selected genes. Further epidemiological investigation of hereditary cancer in dogs is warranted, and should consider not only selected cancers but all cancers occurring in the population. As in humans, it is likely that hereditary genetic risk factors affecting tumor suppressor genes predispose dogs not only to a single cancer type but to several cancer types of both mesenchymal and epithelial origin. Hence, other types of neoplasia, such as mammary neoplasia should be accounted for when evaluating a familial history of cancer within a pedigree. 

## Figures and Tables

**Table 1 vetsci-10-00476-t001:** Table summarizes the described canine GWASs investigating genetic predisposition to sarcomas in dogs. Genetic localization is reported with reference to chromosomal positions in CanFam3.1 (CFA). There are differences in defining the associated regions and the significance between these; hence, this table illustrates the loci and putative candidate genes highlighted in the papers.

Author	Breeds	Sarcoma Type	Major Candidate Loci	Candidate Genes Highlighted in Study
Tonomura et al. [46]	Golden retriever	Hemangiosarcoma	CFA5: 29.6–29.9 Mb CFA5: 33.8–34.1 Mb	*TRPC6*, *STX8*
Karlsson et al. [47]	Irish wolfhound, greyhound, Rottweiler	Osteosarcoma	33 loci identified CFA11: 41.36–41.37 as associated in greyhounds and nearly fixed in Rottweilers and Irish wolfhounds	*CDKN2A/CDKN2B*, *CDKN2B-AS1*
Letko et al. [48]	Leonberger	Osteosarcoma	CFA11: 39.4–42.7 Mb	*CDKN2A/CDKN2B*
Shearing et al. [44]	Bernese mountain dog	Histiocytic sarcoma	CFA11: 36.0–45.8 Mb	*CDKN2A/CDKN2B*, *MTAP*
Evans et al. [49]	Flat-coated retriever	Histiocytic sarcoma	CFA5: 25.0–40.0 Mb CFA19: 50.5–53.0 Mb	*PIK3R6*, *TNFAIP6*
Hedan et al. [45]	Bernese mountain dog, flat-coated retriever, golden retriever, Rottweiler	Histiocytic sarcoma	CFA2:29.15–29.3 CFA5:25.5–34.5 CFA11:29.9–52.4 CFA14:0.5–11.1 CFA20:31.0–32.7 Mb	*CDKN2A/CDKN2B*, *FHIT*, *POT1*, *POT-AS1*, *SPNS3*, *CDKN2B-AS1*, *C9orf72*

## Data Availability

Not applicable.

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
