# Peer review of "Sarcoma Predisposition in Dogs with a Comparative View to Human Orthologous Disease"

_vetsci, 2023, doi:10.3390/vetsci10070476_

Round 1

Reviewer 1 Report

This is an overall well written and informative review. It might benefit from a few statements regarding why this review is needed or timely. The conclusion reached (largely, that more research is needed(?)) is valid, but not an especially strong one.

While overall well written, there are numerous typographical errors and run on sentences (and one non-sentence). The main issue is a distracting over-use of commas. Please see highlights and comments in the accompanying PDF.

Author Response

Dear Reviewer

Thank you for the many useful and constructive comments and suggestions highlighted through the document. We have removed the many incorrectly placed commas and also edited the text throughout the document to make it clearer. We hope that these changes comply with the reviewer’s requests.

Reviewer 2 Report

Line 38 - 40: restructure this sentence.  And include what the previous incidence of sarcomas in humans was. 

Line 46-47.  Restructure this sentence as this is not clear. - what was the % of malignant tumours which were sarcoma in dogs?

Line 48 - 49:  UK used before United Kingdom is spelt out.   Line 48 is also confusing/needs grammatical correction. 

Line 50; remove pr and replace with /.  Can you define the term dog years are risk and how you can compare with the human data?

Line 57-58, 69 - 72, line 122, 141-142, 146, 162. grammatical errors.  Rewrite these sentences. 

Line 79.  space between cancer and type.

Line 132 - 134 ;  provide a reference. 

Line 214; first use of chr instead of chromasome. 

Overall too many commas!  Remove some eg line 99 - this needs no commas.

In general, dog breeds do not need to be capitalised - and you are inconsistent in your text which some breeds given a capital and others not. 

You need to reference Table 1 in your text. 

I would suggest a second table including highlighted human genes associated with sarcoma to show similarities/differences between humans and dogs. 

This manuscript needs minor revision, mainly to remove excessive numbers of commas, as many sentences do not flow well.  

All other comments are included for the author. 

Author Response

Response to reviewer 1

Dear reviewer

Thank you for the constructive comments and suggestions. We have edited the manuscript accordingly and responded to each of the suggestions below.

Line 38 - 40: restructure this sentence.  And include what the previous incidence of sarcomas in humans was. 

Thank you for this comment. We have now re-written this section and it now reads ‘’An epidemiological study investigating the incidence of sarcomas in the US between 2002 and 2014, showed that the overall incidence in the period was 7.1/100.000 people. However when looking at individual years there was a trend showing an increase in incidence during the study period with the incidence raising from 6.8 /100.000 people in 2002 to 7.8/100.000 people in 2014 [7]’’

Line 46-47.  Restructure this sentence as this is not clear. - what was the % of malignant tumours which were sarcoma in dogs?

Thank you for this suggestion. Unfortunately the exact percentage in this study is not possible to calculate the precise percentage due to the nature of the data available from this study. I have tried to make the data more transparent by writing ‘’ A recent German study evaluating the histopathological diagnosis of 70,966 benign and malignant canine tumors, showed that fibrosarcomas alone accounted for 5.8% of the neoplastic lesions however when including other sarcoma types, the overall percentage of sarcomas is closer to 13%, though the lack of histological details in the study does not allow for a precise estimation. If only comparing to malignant tumours in this database, the fraction would likely be substantially larger[8]’’

Line 48 - 49:  UK used before United Kingdom is spelt out.   Line 48 is also confusing/needs grammatical correction. 

Thank you, this has been corrected and it now reads ‘’ A study from the United Kingdom (UK), investigated the yearly incidence of different tumors based on insurance data from dogs’’

Line 50; remove pr and replace with /.  Can you define the term dog years are risk and how you can compare with the human data?

Thank you for this comment we have changed the text in this section to correctly reflect yearly incidence which now makes it more comparable to the human data.  

The yearly incidence of soft tissue sarcomas in the study was 122/100,000 dogs, whilst the incidence for osteosarcoma was 57/100,000 dogs. The incidence for hemangiosarcoma in the study was 24/100,000 dogs whilst the incidence for cardiac hemangiosarcoma was 1/100,000 dogs [9]. It is difficult to directly compare human and canine epidemiological data due to differences in disease surveillance and the accelerated life span of dogs however, overall sarcomas account for a larger fraction of malignant tumors and occur with a substantially higher yearly incidence, in dogs compared to humans.

Line 57-58, 69 - 72, line 122, 141-142, 146, 162. grammatical errors.  Rewrite these sentences. 

Thank you we have addressed these sentences in the text

Line 79.  space between cancer and type.

Thank you, this has been corrected

Line 132 - 134 ;  provide a reference. 

References have been added to this section.

Line 214; first use of chr instead of chromasome. 

Thank you, to be consistent we have corrected the text to only use chromosome and not the abbreviation.

Overall too many commas!  Remove some eg line 99 - this needs no commas.

Thank you, the first author was a bit generous with commas prior to the final submission. We have now substantially reduced the number of commas.

In general, dog breeds do not need to be capitalised - and you are inconsistent in your text which some breeds given a capital and others not. 

Thank you for this comment. I have cross referenced each dog breed in the Oxford dictionary to assure that dog breeds which are nouns are capitalized correctly. I have not followed this rule in table 1 and hence this has now been changed accordingly so only the nouns Rottweiler, Bernese mountain dog, Labrador retriever are capitalized.

You need to reference Table 1 in your text. 

Thank you we have added the sentence ‘’ Major findings from these studies are summarized in table 1’’

I would suggest a second table including highlighted human genes associated with sarcoma to show similarities/differences between humans and dogs. 

Thank you for this suggestion. This was even considered in the first version of the manuscript however due to the major differences in the methods used in the human and canine studies we feel that it would be misleading to try and make a direct comparison. Really additional whole genome sequencing studies will need to be performed to make this comparison between species which the first author is in the process of performing.